# 4D3R: Motion-Aware Neural Reconstruction and Rendering of Dynamic Scenes from Monocular Videos

**Mengqi Guo**[1], **Bo Xu**[2], **Yanyan Li**[1], **Gim Hee Lee**[1]

[1]National University of Singapore
[2]Wuhan University
{mengqi.guo, gimhee.lee}@comp.nus.edu.sg

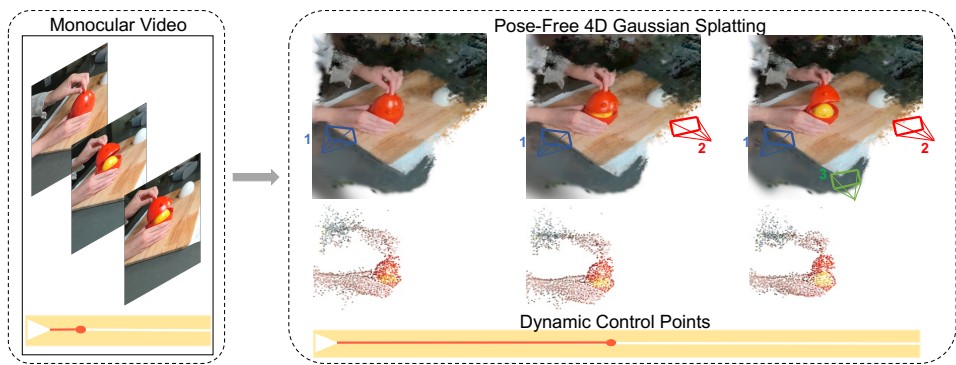

Figure 1: Overview of our pose-free 4D Gaussian Splatting. Given a monocular video sequence of a dynamic scene (left), our method directly reconstructs the 4D scene without pre-computed camera poses (right). Dynamic control points guide the deformation of Gaussian points to model motion, producing high-quality novel views across different time steps.

## Abstract

Novel view synthesis from monocular videos of dynamic scenes with unknown camera poses remains a fundamental challenge in computer vision and graphics. While recent advances in 3D representations such as Neural Radiance Fields (NeRF) and 3D Gaussian Splatting (3DGS) have shown promising results for static scenes, they struggle with dynamic content and typically rely on pre-computed camera poses. We present 4D3R, a pose-free dynamic neural rendering framework that decouples static and dynamic components through a two-stage approach. Our method first leverages 3D foundational models for initial pose and geometry estimation, followed by motion-aware refinement. 4D3R introduces two key technical innovations: (1) a motion-aware bundle adjustment (MA-BA) module that combines transformer-based learned priors with SAM2 for robust dynamic object segmentation, enabling more accurate camera pose refinement; and (2) an efficient Motion-Aware Gaussian Splatting (MA-GS) representation that uses control points with a deformation field MLP and linear blend skinning to model dynamic motion, significantly reducing computational cost while maintaining high-quality reconstruction. Extensive experiments on real-world dynamic datasets demonstrate that our approach achieves up to 1.8dB PSNR improvement over state-of-the-art methods, particularly in challenging scenarios with large dynamic objects, while reducing computational requirements by 5× compared to previous dynamic scene representations.

39th Conference on Neural Information Processing Systems (NeurIPS 2025).

# 1 Introduction

Novel view rendering from monocular videos of dynamic scenes remains a fundamental challenge in both computer graphics and computer vision communities. While static scene reconstruction has seen significant advances with methods like 3D Gaussian Splatting (3DGS) [25, 58, 29] and Neural Radiance Fields (NeRF) [38, 20, 3, 1, 2], dynamic scenes present substantially greater challenges. Unlike static environments, dynamic scenes require modeling intricate 3D environments with temporal coherence while handling complex camera viewpoints. These approaches leverage Gaussian primitives or neural layers but face severe challenges in capturing scene evolution with high fidelity, especially when addressing substantial changes in handling dynamic content such as managing motion, ensuring temporal consistency, and maintaining efficient scene representations.

Adapting 3DGS to dynamic scenes has led to the development of various 4D-GS approaches [60, 22] that incorporate deformation fields modeled by multi-layer perceptrons (MLPs), motion bases, or 4D representations. Despite achieving quantitative improvements, these methods typically rely on pre-computed camera poses from multi-view systems or Structure-from-Motion pipelines, which often fail in scenes with significant dynamic content. This dependency on ground truth or pre-estimated camera poses severely limits their applicability in real-world environments.

Estimation of 6-DoF camera poses typically involves establishing 2D-3D correspondences followed by solving the Perspective-n-Point (PnP) [19] problem with RANSAC [14]. Approaches for predicting 2D-3D correspondences can be broadly categorized into two main directions: Structure-from-Motion (SfM) methods such as COLMAP [47, 48], and scene coordinate regression (SCR) [49]. SfM methods detect and describe key points in 2D images [53, 11], linking them to corresponding 3D coordinates [34, 46]. Although these methods are effective, they still face challenges including high computational overhead, significant storage requirements, and potential privacy concerns when processing sensitive data [51]. In contrast, SCR methods [49, 5, 56, 71] utilize deep neural networks (DNNs) to directly predict the 3D coordinates of the image pixels, followed by running PnP with RANSAC for camera pose estimation. These approaches offer notable advantages such as higher accuracy in smaller scenes, reduced training times, and minimizing storage requirements. Taking advantage of these benefits, this paper adopts SCR over traditional SfM methods for camera pose estimation. Furthermore, DUSt3R [56] employs a Vision Transformer (ViT)-based architecture to predict 3D coordinates using a data-driven approach and in the following work, MonST3R [71] extends DUSt3R to dynamic scenes by fine-tuning the model on suitable dynamic datasets. However, treating pose estimation and scene reconstruction as separate tasks in dynamic environments typically leads to suboptimal performance.

A straightforward approach to addressing the pose-free novel view synthesis problem is to directly combine MonST3R [71] with 4D-GS methods [22]. However, the accuracy of camera poses predicted by MonST3R is not sufficiently stable, causing 4D-GS methods to struggle with reconstructing accurate scenes and resulting in poor rendering quality. Particularly, these methods often fail in scenarios involving moving objects that occupy a significant portion of the image. Since correspondences between 2D keypoints and 3D points are established based on static scene elements, dynamic objects are commonly deemed to be outliers during the RANSAC process. This assumption breaks down in the presence of large or dominant moving objects, further degrading performance in dynamic environments.

To overcome these issues, this paper proposes a novel architecture for pose-free dynamic Gaussian Splatting that integrates transformer-based motion priors for initial pose estimation and then refines it using a Motion-Aware Bundle Adjustment (MA-BA) module. Our key insight is that the motion mask and scene reconstruction should be jointly optimized rather than treated as isolated processes, allowing for more accurate camera pose estimation and higher-quality novel view synthesis. Specifically, the ViT-based transformer gives the initial dynamic mask. We sample the top-K values and turn their location into K-point prompts for pretrained SAM2 [43]. Finally, the final dynamic mask is the combination of the output dynamic object segments from the SAM2 and the confidence map from the transformer. The dynamic mask serves as a static point selection when performing RANSAC, which can reduce the noise introduced by the dynamic points.

For 4D-GS, the expensive computational cost is a huge burden since millions of GS points need to learn a set of motion parameters. Some works design compact representations to solve this problem, such as the sparse motion basis [23], sparse-control points [22], and k-plane [60]. We design our 4D

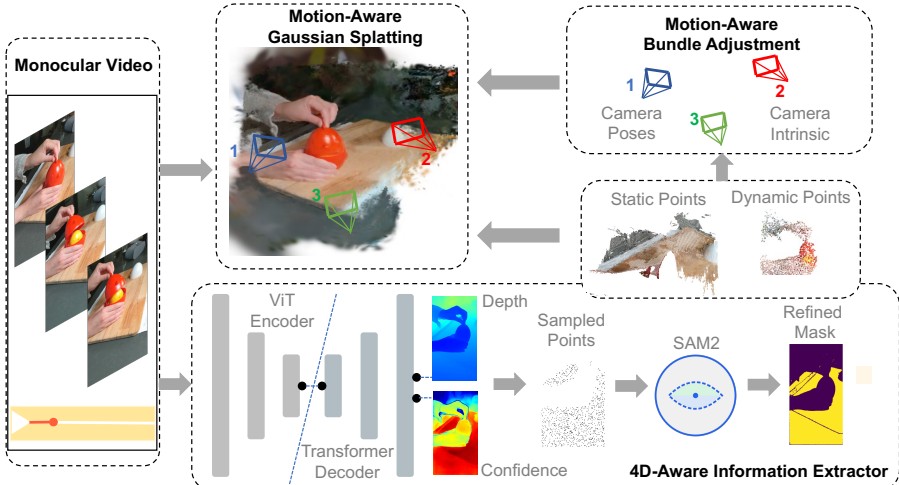

Figure 2: Overview of our motion-aware 4D gaussian splatting pipeline. Our framework consists of three main modules: (1) A 4D-aware information extractor that processes input frames through parallel ViT encoders and decoders to extract geometric and motion information; (2) A motion-aware bundle adjustment module that leverages motion predictions for robust camera estimation; and (3) A motion-aware gaussian splatting module that enables dynamic scene modeling through adaptive control points.

representation of Motion-Aware Gaussian Splatting (MA-GS) based on motion masks generated from the MA-BA module. Specifically, we model the dynamic motion by hundreds of control points with a deformation field MLP, and Gaussian points are using Linear Blend Skinning (LBS). The model first trains the control points for the dynamic part and then trains the GS points.

We show the effectiveness of our approach through extensive experiments on both synthetic and real-world dynamic scenes. Our findings demonstrate notable gains in pose estimation accuracy and reconstruction quality over current methods. In particular, we outperform state-of-the-art techniques on difficult dynamic scenes by achieving improvements of 1.8 in PSNR in novel view rendering quality and more accurate pose estimation.

Our key contributions include: 1) We propose a novel motion-aware pipeline that fundamentally integrates pose estimation with scene reconstruction, eliminating the traditional separation that causes failures in highly dynamic scenes. 2) We introduce a theoretical framework for motion-aware bundle adjustment that jointly optimizes camera poses and scene representation, enabling robust performance even when moving objects dominate the scene. 3) We design a compact and efficient 4D representation using motion-aware gaussian splatting that significantly reduces memory requirements while maintaining rendering quality. 4) Our approach demonstrates state-of-the-art performance on challenging dynamic scenes without requiring pre-computed poses, enabling truly monocular novel view synthesis.

## 2    Related Work

### 2.1    Static Scene Novel View Rendering.

Novel View Synthesis aims to generate novel viewpoints from a set of observations. Recently, neural implicit representations have shown impressive capabilities. NeRF [38] achieved breakthrough results by representing scenes through MLPs. Subsequent work focused on acceleration through methods like baking [21] and explicit representations [39]. 3DGS [25] introduced efficient rasterization of anisotropic 3D Gaussians, enabling real-time rendering without quality degradation. Recent extensions have ireal-time rendering [68, 44], camera modeling [61], faster training [35, 16, 39, 7], and sparse view [69, 45]. However, these methods assume static scenes and known camera parameters, limiting their practical applications.

## 2.2 Dynamic Scene Novel View Rendering.

Research has expanded to capture both motion and geometry in dynamic scenes [64, 37, 66, 31, 15, 32, 30]. Initial approaches [42, 40] learned additional time-varying deformation fields. Alternative methods [28, 16, 62] encode scene dynamics through multi-dimensional feature fields without explicit motion modeling. Following 3DGS, recent work [60] proposes learning individual Gaussian trajectories over time. More efficient representations have emerged, including factorized motion bases [27] and sparse control points [22]. Another approach [65] extends spherical harmonics to 4D. As noted in Dycheck [18], many methods focus on unrealistic scenarios, while real-world capture involves substantial motion. To resolve motion ambiguity, recent work leverages pretrained depth estimation [63] or trajectory tracking [24]. Our approach utilizes DUSt3R [56] for initialization and incorporates SAM2 [43] for dynamic motion segmentation.

## 2.3 Pose-Free Neural Field.

Traditional NVS relies heavily on SfM [47] for camera parameters. Recent research explores optimizing neural fields without pre-calibrated poses. iNeRF[67] estimates camera poses from pre-trained NeRF through photometric optimization. NeRF– [59] jointly optimize camera and scene parameters with geometric regularization. BARF [33] and GARF [10] address gradient inconsistency in positional embeddings. Nope-NeRF [4] leverages geometric priors for accurate camera estimation. In the 3DGS domain, CF3DGS [17] introduces progressive optimization, while InstantSplat [12] uses DUSt3R [56] for initialization but remains limited to static scenes. ZeroGS [9] utilizes the DUSt3R and progressive image registration for pose-free 3D GS. Our approach differs by introducing a pose-free pipeline for dynamic scenes that decouples static backgrounds from dynamic objects. We utilize DUSt3R's geometric foundation model and leverage 3DGS's explicit nature for enhanced geometric regularization.

## 3 Method

### 3.1 Preliminaries

3D Gaussian splatting represents scenes using a collection of colored 3D Gaussian primitives. Each Gaussian $G_j$ is characterized by its center position $\boldsymbol{\mu}_j$, covariance matrix $\boldsymbol{\Sigma}_j$ (parameterized by rotation quaternion $\mathbf{q}_j$ and scaling vector $\mathbf{s}_j$), opacity value $\sigma_j$, and spherical harmonic coefficients $\mathbf{sh}_j$ for view-dependent appearance. The scene representation is thus $\mathcal{G} = \{G_j : \boldsymbol{\mu}_j, \mathbf{q}_j, \mathbf{s}_j, \sigma_j, \mathbf{sh}_j\}$.

During rendering, these 3D Gaussians are projected onto the image plane with transformed 2D covariance matrices $\boldsymbol{\Sigma}'$. The final color at each pixel is computed through $\alpha$-blending:

$$C(\mathbf{u}) = \sum_{i \in \mathcal{N}} T_i \alpha_i \, \text{SH}(\mathbf{sh}_i, \mathbf{v}_i), \quad \text{where } T_i = \prod_{j=1}^{i-1}(1 - \alpha_j) \tag{1}$$

Our framework extends Gaussian Splatting to dynamic scenes with sparse control while maintaining computational efficiency and rendering quality. For more details, please refer to the supplementary.

### 3.2 Problem Definition and Overview

Given a monocular video sequence $\mathcal{V} = \{I_t\}_{t=1}^{\top}$ capturing a dynamic scene with moving objects and camera motion, our goal is to reconstruct a complete 4D representation of the scene. Our pipeline estimates the following parameters: 1) Camera parameters $\mathcal{C}_t = \{K_t, R_t, T_t\}$, where $K_t \in \mathbb{R}^{3 \times 3}$ represents the intrinsic matrix, and $[R_t \mid T_t] \in SE(3)$ denotes the extrinsic parameters. 2) Dense depth map $D_t \in \mathbb{R}^{H \times W}$ and motion map $M_t \in \{0, 1\}^{H \times W}$ capturing per-pixel information. 3) Dynamic scene representation through motion-aware Gaussian Splatting parameters $\mathcal{G}$, motion-aware control points $\mathbb{P}$, and a deformation field MLP $\Theta$.

As illustrated in Fig 2, our framework combines implicit geometric estimation and explicit motion understanding to address the unique challenges of dynamic scene reconstruction from monocular video through three primary components: 4D-aware information extractor, Motion-Aware Bundle Adjustment (MA-BA) and Motion-Aware Gaussian Splatting (MA-GS) representation.

### 3.3  4D-Aware Information Extraction

Our 4D-aware information extractor serves as the foundation for both camera pose estimation and scene reconstruction by leveraging pre-trained vision models to extract geometric and motion information from monocular inputs. To have a good initialization of the Gaussian splatting, we first employ a pre-trained ViT-based transformer model from MonST3R [71] that sequentially processes each input frame $I_t$ through an encoder-decoder block that extracts deep features to give a scene coordinate map $X_t \in \mathbb{R}^{H \times W \times 3}$ representing the 3D structure and a confidence map $W_t \in \mathbb{R}^{H \times W}$ indicating the reliability of the predictions. We also include the optical flow from SEA-RAFT [57].

Using the scene coordinate map $X_t$ and confidence map $W_t$, we obtain high-confidence points $\mathcal{S}$ through a principled two-step filtering process:

$$\mathcal{S} = \{p_i \mid W_t(p_i) > \tau_c \wedge D_t(p_i) < \tau_d\}, \tag{2}$$

where $\tau_c$ and $\tau_d$ are the confidence and depth thresholds, respectively. This filtering strategy is based on two key insights: 1) The selection of points with high confidence scores are more likely to yield reliable geometric estimates. 2) Filtering out points at infinity with zero disparity that give unreliable depth estimates.

Unlike previous approaches that rely solely on motion estimators, we leverage SAM2 [43] semantic understanding ability to generate pixel-precise dynamic object segmentation $M_t \in \{0, 1\}^{H \times W}$, with high-confidence points $\mathcal{S}$ as prompts. This method critically enables our motion-aware processing pipeline to handle scenes dominated by dynamic content.

### 3.4  Motion-Aware Bundle Adjustment

Our MA-BA module introduces an approach to camera pose estimation that explicitly models the separation between static and dynamic scene components, addressing a fundamental limitation in traditional bundle adjustment methods. We leverage the dynamic region mask $M_t$ to enhance the accuracy of camera pose estimation. For frame pairs $(I_t, I_{t'})$, we introduce a masked PnP-RANSAC approach that focuses solely on static regions:

$$\mathcal{P}_{static} = \{p_i \in \mathcal{S} \mid M_t(p_i) = 0\}. \tag{3}$$

By restricting correspondence matching to static regions, we significantly reduce the likelihood of incorrect matches due to dynamic objects. The optimization objective becomes:

$$E(R_t, T_t) = \sum_{p_i \in \mathcal{P}_{static}} \|\Pi(R_t p_i + T_t) - p_i'\|^2. \tag{4}$$

where $\Pi(\cdot)$ is the camera model mapping a set of 3D points onto the image.

We further refine the camera poses through Differentiable Dense Bundle Adjustment (DBA) layer [54]. For more details, please refer to the supplementary.

### 3.5  Motion-Aware Gaussian Splatting

Our Motion-Aware Gaussian Splatting (MA-GS) module introduces a principled approach to dynamic scene representation that significantly reduces the parameter space by focusing computational resources on regions requiring deformation modeling. We adopt a set of control points $\mathcal{P} = (p_i \in \mathbb{R}^3, \sigma_i \in \mathbb{R}^+)_{i=1}^{N_p}$, where $p_i$ represents the 3D coordinate in the canonical space and $\sigma_i$ defines the radius of the Radial Basis Function (RBF) kernel. These control points are initialized from our motion map $M_t$, where the static and dynamic regions are handled distinctly through our MA-GS module.

In the first stage, we optimize the control points on the dynamic regions with the following loss:

$$L_{control} = \sum_{p_i \in \mathcal{P}} M_t(p_i) L_{render}(p_i), \tag{5}$$

where $M_t(p_i)$ acts as a binary mask to selectively optimize only the control points in dynamic regions and $L_{render}$ refers to the standard photometric loss between rendered and ground truth pixels, using L1 and DSSIM metrics. The equation selectively applies this loss only to dynamic regions via the

Mt(pi) binary mask. This targeted optimization significantly reduces the complexity by focusing only on regions that require deformation modeling. For dynamic control points, we learn time-varying transformations through a specialized mapping function:

$$\Theta : (p_t, t) \rightarrow (R_t^k, T_t^k), \tag{6}$$

where $[R_t^k \mid T_t^k] \in SE(3)$ represents the six-degrees-of-freedom transformation at time $t$, $\Theta$ is the deformation field MLP. For numerical stability and continuous interpolation, we represent rotations using unit quaternions $r_t^k \in \mathbb{H}$, where $\mathbb{H}$ denotes the space of unit quaternions. The dynamic scene rendering process utilizes these control points through weighted transformation blending.

In the second stage, we optimize the Gaussian parameters $G_j = \{\boldsymbol{\mu}_j, \mathbf{q}_j, \mathbf{s}_j, \sigma_j, \mathbf{sh}_j\}$ by applying motion-aware constraints through a detached gradient path:

$$\mu_j' = \begin{cases} \mu_j & \text{if } M_t(\mu_j) = 0 \\ \sum_{k \in \mathcal{N}_j} w_{jk}(R_t^k(\mu_j - p_k) + p_k + T_t^k) & \text{otherwise}, \end{cases} \tag{7}$$

where $w_{ij}$ is Linear Blend Skinning (LBS) weight [52]. Crucially, this constraint is implemented with gradient detachment to ensure the motion-aware transformation does not affect the parameter updates of the MLP and vice versa. This prevents competing optimization objectives between Gaussian parameters and deformation field parameters, leading to more stable convergence and better results. The LBS weights are computed with a normalized exponential kernel:

$$w_{jk} = \tilde{w}_{jk} / \sum_{k \in \mathcal{N}j} \tilde{w}_{jk}, \quad \text{where } \tilde{w}_{jk} = \exp(-d_{jk}^2 / 2\sigma_k^2). \tag{8}$$

$d_{jk}$ represents the Euclidean distance between Gaussian center $\mu_j$ and control point $p_k$, and $\mathcal{N}_j$ denotes the set of $K$-nearest neighboring control points for Gaussian $j$.

For orientation updates, we employ quaternion blending to ensure smooth rotational deformation:

$$q_j' = \left( \sum_{k \in \mathcal{N}j} w_{jk} r_t^k \right) \otimes q_j, \tag{9}$$

where $\otimes$ denotes quaternion multiplication. This formulation ensures the entire dynamic scene experiences smooth and physically realistic deformations while maintaining computational efficiency through our motion-aware design.

## 3.6 Training Strategy

As mentioned in the previous section, our training process optimizes the entire scene representation through the two-stage procedure. Specifically, we optimize for the control points in the first stage with $L_{control}$. The second stage of optimization for the motion-aware Gaussian parameters is achieved in the rendering loss $L_{render}$ using L1 distance and DSSIM metrics as follows:

$$L = L_{render} + \lambda_{arap} L_{arap} + \lambda_{rigid} L_{rigid}, \tag{10}$$

where $L_{arap}$ enforces local rigidity with as-rigid-as-possible regularization [50]:

$$L_{arap}(p_i, t_1, t_2) = \sum w_{ik} \| (p_i^{t_1} - p_k^{t_1}) - R_i(p_i^{t_2} - p_k^{t_2}) \|^2, \tag{11}$$

and $L_{rigid}$ enforces rigidity in static regions:

$$L_{rigid} = \sum_j (1 - M_t(\mu_j)) \| \mu_j' - \mu_j \|^2. \tag{12}$$

We employ an adaptive control point strategy that optimizes point distribution based on reconstruction impact. We compute the gradient magnitude of the rendering loss with respect to Gaussian positions, weighted by their influence radius:

$$g_k = \sum \tilde{w}_j \left\| \frac{\partial L}{\partial \mu_j} \right\|^2, \tag{13}$$

where $\tilde{w}_j$ represents the LBS weights connecting Gaussian points to control points, while $\frac{\partial L}{\partial \mu_j}$ is the gradient of the loss with respect to Gaussian positions. This gradient magnitude sum ($g_k$) measures each control point's influence on reconstruction quality. During optimization, we add control points in areas with high $g_k$, adaptively refining the representation where needed most. This adaptive approach ensures effective representation while maintaining high reconstruction quality across diverse dynamic scenes with varying motion complexity.

Table 1: Quantitative results on HyperNeRF's [41] dataset. The best and the second best results are denoted by pink and yellow. The rendering resolution is 960x540. "Time" in the table stands for training times plus camera pose estimation time.

| Model | COLMAP | PSNR(dB)↑ | MS-SSIM↑ | Times↓ | FPS↑ | Storage(MB)↓ |
|---|---|---|---|---|---|---|
| Nerfies [40] | ✓ | 22.2 | 0.803 | ∼ hours | < 1 | - |
| HyperNeRF [41] | ✓ | 22.4 | 0.814 | 32 hours | < 1 | - |
| TiNeuVox-B [13] | ✓ | 24.3 | 0.836 | 3.5 hours | 1 | 48 |
| 3D-GS [25] | ✓ | 19.7 | 0.680 | 4 hours | 55 | 52 |
| FDNeRF [70] | ✓ | 24.2 | 0.842 | - | 0.05 | 440 |
| 4DGS [60] | ✓ | 25.2 | 0.845 | 5 hours | 34 | 61 |
| SC-GS [22] | ✓ | 25.3 | 0.841 | 4 hours | 45 | 85 |
| MonST3R+SC-GS | ✗ | 20.4 | 0.697 | 2 hours | 45 | 153 |
| RoDynRF [36] | ✗ | 23.8 | 0.820 | 28 hours | <1 | 200 |
| Ours | ✗ | 25.6 | 0.844 | 50 mins | 45 | 80 |

Table 2: Quantitative results on DyNeRF's [28] dataset. The best and the second best results are denoted by pink and yellow. "Time" in the table stands for training times plus pose estimation time.

| Model | COLMAP | PSNR(dB)↑ | MS-SSIM↑ | Times↓ | FPS↑ | Storage(MB)↓ |
|---|---|---|---|---|---|---|
| HyperNeRF [41] | ✓ | 16.9 | 0.638 | 32 hours | < 1 | - |
| TiNeuVox-B [13] | ✓ | 18.2 | 0.712 | 3.5 hours | 1 | 48 |
| 3D-GS [25] | ✓ | 15.3 | 0.541 | 4 hours | 55 | 52 |
| 4DGS [60] | ✓ | 18.9 | 0.740 | 5 hours | 34 | 61 |
| SC-GS [22] | ✓ | 19.0 | 0.746 | 4 hours | 45 | 85 |
| MonST3R+SC-GS | ✗ | 16.4 | 0.624 | 2 hours | 45 | 153 |
| RoDynRF [36] | ✗ | 17.8 | 0.620 | 28 hours | <1 | 200 |
| Ours | ✗ | 19.6 | 0.755 | 50 mins | 45 | 80 |

## 4 Experiments

### 4.1 Experimental Settings

**Datasets.** We evaluate our approach on three representative datasets: HyperNeRF's dataset [41], DyNeRF dataset [28], and the MPI Sintel dataset [6]. The DyNeRF dataset features controlled dynamic scenes captured with synchronized cameras, offering a solid baseline for evaluation. We use only one camera view for training, treating it as a monocular sequence. The HyperNeRF dataset presents more challenging scenarios with complex object deformations and camera movements. The MPI Sintel dataset provides ground truth camera poses, enabling quantitative evaluation of our pose estimation accuracy.

**Evaluation Metrics** We evaluate using standard metrics for Novel View Synthesis: Peak Signal-to-Noise Ratio (PSNR), Structural Similarity (SSIM), and Multi-Scale SSIM (MS-SSIM). For camera pose estimation, we report the same metrics as [8]: Absolute Translation Error (ATE), Relative Translation Error (RPE trans), and Relative Rotation Error (RPE rot), after applying a Sim(3) Umeyama alignment on prediction to the ground truth.

**Baselines.** For a comprehensive evaluation, we compare our approach against state-of-the-art methods in both novel view rendering and pose estimation. For novel view rendering, we consider methods designed for static scenes like 3DGS [25], and dynamic scene methods including Nerfies [40], HyperNeRF [41], TiNeuVox [13], 4DGS [60], FDNeRF [70], and SC-GS [22], which represent the current state-of-the-art in dynamic scene modeling. We further compare with pose-free 4D novel view rendering baselines RoDynRF [36] and a strong baseline combining MonST3R [71] (one of the best dynamic pose estimation methods) with SC-GS [22] (one of the best dynamic scene modeling methods). For pose estimation evaluation, we compare against established methods such as DROID-SLAM [54], DPVO [55], and ParticleSFM [73], noting that these methods require camera intrinsics as input.

**Implementation Details.** Our implementation uses a ViT-based transformer for 4D information extraction, pre-trained on DUSt3R [56] and fine-tuned on MonST3R [71] datasets. For dynamic segmentation refinement, we employ SAM2 [43] with multi-point prompting. The Motion-Aware

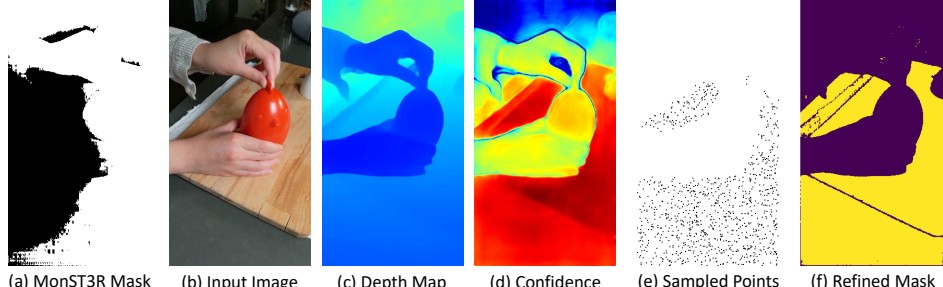

| (a) MonST3R Mask | (b) Input Image | (c) Depth Map | (d) Confidence | (e) Sampled Points | (f) Refined Mask |

Figure 3: Our motion mask refinement pipeline: (a) Initial dynamic mask from MonST3R showing coarse segmentation, (b) Input image of tomato cutting scene, (c) Estimated depth map highlighting object boundaries, (d) Confidence map indicating regions of dynamic motion, (e) Strategically sampled points for mask refinement, and (f) Final refined mask after SAM2 processing showing improved object boundary delineation. The pipeline effectively captures the dynamic nature of the cutting motion while maintaining precise object boundaries.

Table 3: Ablation study of our proposed modules on HyperNeRF dataset.

| Method | PSNR(dB)↑ | MS-SSIM↑ | Times↓ |
|---|---|---|---|
| w/o motion-map | 20.4 | 0.697 | 2 hours |
| w/o SAM-refine | 23.8 | 0.765 | 1 hours |
| w/o MA-GS | 23.5 | 0.734 | 1.5 hours |
| Ours | 25.6 | 0.844 | 50mins |

Gaussian Splatting uses 512 control points, optimized using Adam with learning rates from 1e-4 to 1e-7 (exponential decay). All experiments run on a single NVIDIA RTX3090 GPU.

## 4.2 Results on Novel View Rendering

A key advantage of our approach is its superior performance on challenging scenes with dynamic objects. On the DyNeRF dataset (Tab 2), we achieve state-of-the-art results with a PSNR of 19.6dB and MS-SSIM of 0.775, outperforming existing methods regardless of their reliance on known camera poses. This superior performance comes from our motion-aware components, which effectively handle dynamic objects while maintaining accurate camera pose estimation.

Our method demonstrates exceptional computational efficiency, achieving 5× faster training time compared to COLMAP-dependent methods while maintaining comparable quality. On the HyperNeRF dataset (Tab 1 and Fig 4), we achieve results (PSNR of 25.6dB and MS-SSIM of 0.844) competitive with SC-GS (25.3dB/0.841) and 4DGS (25.2dB/0.845), while significantly outpacing other COLMAP-free approaches like RoDynRF (23.8dB/0.820) and MonST3R+SC-GS (20.4dB/0.697).

Furthermore, our approach excels in resource utilization, maintaining a competitive 45 FPS during inference while requiring only 80MB of memory. This is substantially more efficient than competing methods such as MonST3R+SC-GS (153MB) and RoDynRF (200MB). The efficiency stems from our compact motion-aware representation and efficient motion mask generation pipeline, as illustrated in Fig 2.

## 5 Ablation Studies

To validate the effectiveness of our key components, we conduct comprehensive ablation studies shown in Tab 3:

**Motion-aware Map** Removing the motion-aware map leads to a significant performance drop of 5.2dB in terms of PSNR. This substantial drop confirms our theoretical insight that accurate dynamic-static decomposition is fundamental for handling scenes with large moving objects, addressing the limitation of previous methods like MonST3R that assume dynamic objects occupy only a small portion of the scene.

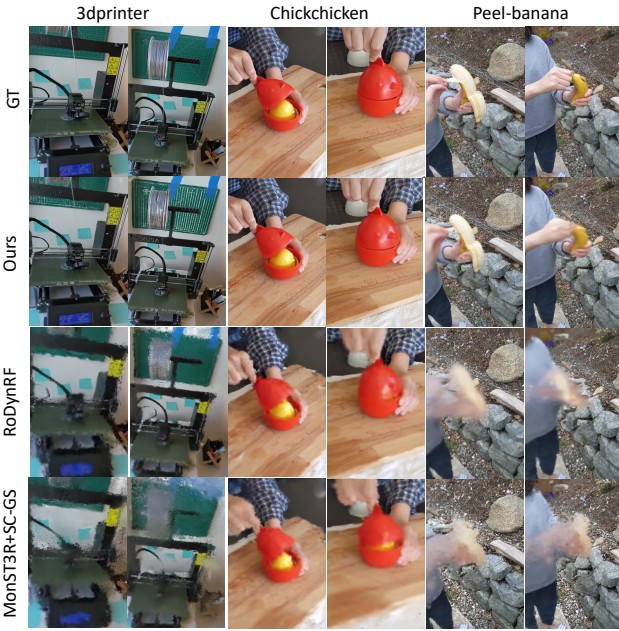

Figure 4: Qualitative comparison with baselines.

**SAM-based Refinement** Without our SAM2-based refinement module, performance decreases to 23.8dB/0.765. The 1.8dB performance gap validates our hypothesis that combining transformer-based motion priors with foundation model segmentation creates a synergistic effect, as shown in Fig 3. The refined masks enable more reliable static point selection during RANSAC, reducing noise from dynamic regions.

**Motion-aware Gaussian Splatting (MA-GS)** Excluding MA-GS results in degraded performance (23.5dB/0.734) while increasing training time by 2x. These results confirm our theoretical prediction that focusing computational resources on motion-significant regions through our control point mechanism substantially improves both efficiency and quality. The improved efficiency (50 mins vs 1.5 hours) demonstrates that our compact representation successfully reduces the search space compared to methods requiring optimization of motion parameters for all Gaussian points.

## 6 Limitation and Broader Impact

Despite our method's improvements, limitations exist: it requires textured frames with sufficient static regions, assumes distinguishable dynamic objects, and struggles with complex non-rigid deformations. While enabling positive applications in AR/VR and remote collaboration, potential misuse exists in surveillance or unauthorized 3D reconstruction. We recommend consent mechanisms for human-centric applications and privacy-preserving rendering techniques. Future work should explore multi-modal sensing, self-supervised segmentation, and privacy-aware reconstruction protocols.

## 7 Conclusion

We presented a novel motion-aware framework for pose-free dynamic novel view synthesis from monocular videos. Our method integrates three key innovations: a 4D-aware information extractor leveraging pre-trained foundation models, a motion-aware bundle adjustment module for robust camera pose estimation, and a compact motion-aware Gaussian splatting representation. Extensive experiments show that our method significantly reduces computational overhead while achieving state-of-the-art performance in pose estimation accuracy and novel view synthesis quality. Compared to COLMAP-dependent methods, our approach achieves 5x faster training times and outperforms existing methods by 1.8dB in PSNR. For more intricate dynamic scenes, future research might investigate adding temporal consistency constraints. Furthermore, examining self-supervised learning strategies for motion mask creation may help lessen dependency on pre-trained models while preserving strong performance.

## Acknowledgment

This research / project is supported by the National Research Foundation (NRF) Singapore, under its NRF-Investigatorship Programme (Award ID. NRF-NRFI09-0008), and the Tier 2 grant MOE-T2EP20124-0015 from the Singapore Ministry of Education.

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

# A Preliminaries

The representation of 3D scenes with Gaussian splatting employs a collection of colored 3D Gaussian primitives. Each Gaussian primitive $G$ is characterized by its center $\boldsymbol{\mu}$ in 3D space and a corresponding 3D covariance matrix $\boldsymbol{\Sigma}$, conforming to:

$$G(\mathbf{x}) = \exp\left(-\frac{1}{2}(\mathbf{x} - \boldsymbol{\mu})^\top \boldsymbol{\Sigma}^{-1}(\mathbf{x} - \boldsymbol{\mu})\right). \tag{14}$$

To facilitate optimization, we decompose the covariance matrix $\boldsymbol{\Sigma}$ into $\mathbf{R}\mathbf{S}\mathbf{S}^\top\mathbf{R}^\top$, where $\mathbf{R}$ represents a rotation matrix encoded by a quaternion $\mathbf{q} \in SO(3)$, and $\mathbf{S}$ denotes a scaling matrix parameterized by a 3D vector $\mathbf{s}$. The complete parameterization of each Gaussian includes an opacity value $\sigma$ governing its rendering influence, alongside spherical harmonic (SH) coefficients $\mathbf{sh}$ that capture view-dependent appearance variations.

The scene representation therefore consists of a set $\mathcal{G} = \{G_j : \boldsymbol{\mu}_j, \mathbf{q}_j, \mathbf{s}_j, \sigma_j, \mathbf{sh}_j\}$, where $\boldsymbol{\mu}_j$ is the position, $\mathbf{q}_j$ is the orientation, $\mathbf{s}_j$ is the scale, $\sigma_j$ is the standard deviation, and $\mathbf{sh}_j$ is the spherical harmonics coefficients. The rendering pipeline projects these 3D Gaussians onto the image plane, where they undergo efficient $\alpha$-blending. During projection, the 2D covariance matrix $\boldsymbol{\Sigma}'$ and center $\boldsymbol{\mu}'$ are computed as:

$$\boldsymbol{\Sigma}' = \mathbf{J}\mathbf{W}\boldsymbol{\Sigma}\mathbf{W}^\top\mathbf{J}^\top, \quad \boldsymbol{\mu}' = JW\boldsymbol{\mu}, \tag{15}$$

where $\mathbf{J}$ is the Jacobian matrix of the linear approximation of the projective transformation and $\mathbf{W}$ is the rotation matrix of the viewpoint. The final color $C(\mathbf{u})$ at pixel $\mathbf{u}$ emerges from neural point-based $\alpha$-blending:

$$C(\mathbf{u}) = \sum_{i \in \mathcal{N}} T_i \alpha_i \, \mathrm{SH}(\mathbf{sh}_i, \mathbf{v}_i), \quad \text{where } T_i = \prod_{j=1}^{i-1}(1 - \alpha_j). \tag{16}$$

Here, $\mathcal{N}$ shows the number of Gaussians that overlap with the pixel $\mathbf{u}$. In this formulation, $\mathrm{SH}(\cdot, \cdot)$ represents the spherical harmonic function evaluated with respect to the view-direction $\mathbf{v}_i$. The $\alpha$-value for each Gaussian is determined by:

$$\alpha_i = \sigma_i \exp\left(-\frac{1}{2}(\mathbf{p} - \boldsymbol{\mu}'_i)^\top \boldsymbol{\Sigma}'^{-1}_i(\mathbf{p} - \boldsymbol{\mu}'_i)\right), \tag{17}$$

where $\boldsymbol{\mu}'_i$ and $\boldsymbol{\Sigma}'_i$ correspond to the projected center and covariance matrix of Gaussian $G_i$. Real-time and high-fidelity image synthesis is achieved through the optimization of Gaussian parameters $\{G_j : \boldsymbol{\mu}_j, \mathbf{q}_j, \mathbf{s}_j, \sigma_j, \mathbf{sh}_j\}$ coupled with adaptive density adjustment. We propose a framework that builds on the foundation of Gaussian Splatting for dynamic scenes by adding sparse control, without compromising computational efficiency and rendering quality.

# B Differentiable Dense Bundle Adjustment (DBA) layer

We further refine the camera poses through the Differentiable Dense Bundle Adjustment (DBA) layer [54], which incorporates optical flow information to improve geometry estimation. This approach is particularly effective in dynamic scenes as it allows us to focus optimization on static regions while accounting for motion consistency in dynamic areas:

$$E_{DBA}(C'_t, d'_t) = \sum_{(i,j) \in \mathcal{E}} (1 - M_t(i))\|p^*_{ij} - \Pi_e(C'_{ij} \circ \Pi_e^{-1}(p_i, d'_i))\|^2_{\Sigma_{ij}}, \tag{18}$$

where $C'_t$ is the camera pose and $d'_t$ is the depth values. $\|\cdot\|_{\Sigma_{ij}}$ is the Mahalanobis distance weighted by the confidence scores, $(i,j) \in \mathcal{E}$ denotes an overlapping field-of-view with shared points between image $I_i$ and $I_j$, $p^*_{ij}$ is the sum of optical flow $r_{ij}$ and $p_{ij}$, and the term $(1 - M_t(i))$ ensures that only static regions contribute to the optimization.

The system optimizes for updated camera pose $C'_t$ and depth values $d'_t$ through a sparse matrix formulation $\Delta\xi_t$ and $\Delta d_t$, which is the normal equation derived from the cost function in Eqn.( 18):

$$\begin{bmatrix} B & E \\ E^\top & C \end{bmatrix} \begin{bmatrix} \Delta\xi_t \\ \Delta d_t \end{bmatrix} = \begin{bmatrix} v \\ w \end{bmatrix} \tag{19}$$

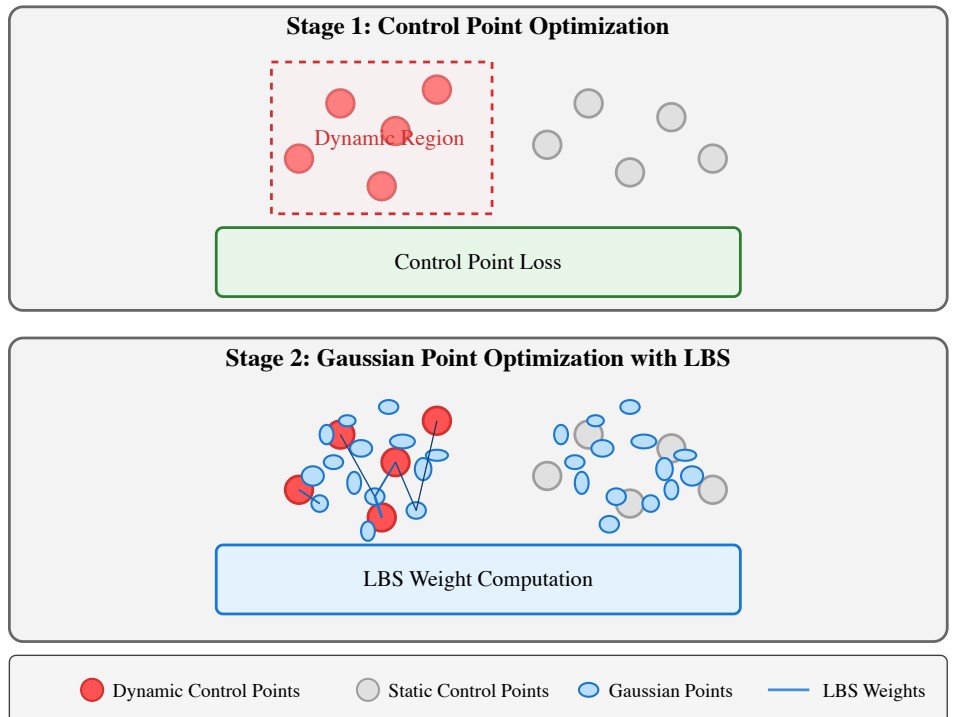

Figure 5: Two-Stage Optimization Process for Motion-Aware Gaussian Splatting. Stage 1 optimizes control points in dynamic regions (red) using control point loss, while static regions (gray) remain fixed. Stage 2 performs Gaussian optimization (blue ellipses) through Linear Blend Skinning, with connection lines showing influence weights between control points and Gaussians.

where $E$ models the coupling between pose and depth parameters, $C$ captures the depth-depth relationships, $B$ represents the pose-pose interactions, $v$ and $w$ are the gradient terms corresponding to pose updates and depth updates, respectively.

## C    Two-stage Optimization Strategy in MAGS

Our two-stage optimization strategy reduces computational cost by focusing on dynamic regions only, as described in Sec. 3.5 of the main paper. We illustrate the strategy in Fig 5, which demonstrates our approach to efficient motion-aware scene reconstruction. In the first stage, we selectively optimize control points only within regions identified as dynamic by our motion segmentation module, significantly reducing the computational burden compared to methods that optimize all scene parameters simultaneously. The dynamic region boundary (indicated by the dashed red outline) separates areas requiring deformation modeling from static background elements. During the second stage, Gaussian primitives are optimized using Linear Blend Skinning weights computed through normalized exponential kernels based on spatial proximity to control points. The varying opacity of connection lines visualizes the influence magnitude of each control point on nearby Gaussian primitives, with stronger connections (darker lines) indicating higher LBS weights. This strategic separation of optimization stages enables our method to achieve both computational efficiency and high-quality reconstruction by concentrating computational resources where motion occurs while maintaining stable anchoring in static regions.

Table 4: Quantitative evaluation of camera poses estimation on the MPI Sintel dataset. The `best` and the `second best` results are denoted by pink and yellow. The methods of the top block discard the dynamic components and do not reconstruct the dynamic scenes; thus they cannot render novel views. We exclude the COLMAP results since it fails to produce poses in 5 out of 14 sequences.

| Method | ATE↓ | RPE trans↓ | RPE rot↓ |
|---|---|---|---|
| DROID-SLAM* [54] | 0.175 | 0.084 | 1.912 |
| DPVO* [55] | 0.115 | 0.072 | 1.975 |
| ParticleSFM [73] | 0.129 | 0.031 | 0.535 |
| LEAP-VO* [8] | 0.089 | 0.066 | 1.250 |
| Robust-CVD [26] | 0.360 | 0.154 | 3.443 |
| CasualSAM [72] | 0.141 | 0.035 | 0.615 |
| DUSt3R [56] w/ mask | 0.417 | 0.250 | 5.796 |
| MonST3R [71] | 0.108 | 0.042 | 0.732 |
| NeRF– [67] | 0.433 | 0.220 | 3.088 |
| BARF [33] | 0.447 | 0.203 | 6.353 |
| RoDynRF [36] | 0.089 | 0.073 | 1.313 |
| Ours | 0.086 | 0.035 | 0.639 |

* requires ground truth camera intrinsics as input

## D  LBS parameter learning

As shown in Eqn. (8) of the main paper, the LBS weights are computed with a normalized exponential kernel. Our motion-aware framework significantly improves the efficiency and stability of LBS parameter learning through three key mechanisms:

1) Focusing control point optimization exclusively on dynamic regions identified by our motion mask, which concentrates computational resources where they're most needed.

2) Preserving static points' positions during the GS optimization process, which provides stable anchors for the scene representation.

3) Substantially reducing the number of points requiring LBS parameter learning, which improves both computational efficiency and optimization stability.

## E  Results on Pose Estimation

Tab 4 presents our pose estimation results on the MPI Sintel dataset, where our method demonstrates exceptional performance across all metrics. The methods in the upper portion of the table discard dynamic components and cannot render novel views. COLMAP results are excluded because it fails to produce poses in 5 out of 14 sequences, which further illustrates the challenges faced by COLMAP-dependent methods in handling general scenes with complex camera motions. We achieved an ATE of 0.086, showing comparable accuracy to the SOTA method LEAP-VO (0.089) and significantly outperforming methods like BARF (0.447) and NeRF– (0.433). For relative pose errors, our method achieves 0.035 for translation and 0.639 for rotation, matching or exceeding the performance of specialized pose estimation methods.

The strong pose estimation performance can be attributed to several factors. First, our MA-BA effectively leverages the dynamic masks refined by our mask refinement pipeline, reducing the noise introduced by dynamic objects during pose optimization. Second, integrating SAM2 for mask refinement significantly improves the accuracy of dynamic object segmentation, leading to more reliable static point selection for RANSAC-based pose estimation.

## F  Technical Discussion and Analysis

### F.1  Technical Contributions

Our motion-aware framework delivers three key innovations: (1) an integrated MA-BA module that combines transformer-based motion priors with SAM2's segmentation capabilities in a unified

pipeline; (2) a gradient-detached dynamic control point mechanism that strategically allocates computational resources to motion-significant regions; and (3) an end-to-end pose-free approach that eliminates dependency on pre-computed camera poses. Empirical validation confirms these innovations deliver substantial improvements, with a 1.3dB PSNR gain over combined baseline components.

### F.2 Scene Coordinate Regression Advantages

Our approach leverages Scene Coordinate Regression (SCR) over Structure from Motion (SfM) for its dual benefits: SCR not only delivers faster and more accurate results but also generates high-quality dynamic masks through our motion-aware bundle adjustment module. This creates a synergistic pipeline where dynamic region identification directly informs reconstruction, enabling more efficient processing while maintaining or exceeding the accuracy of traditional SfM approaches.

### F.3 Two-Stage Optimization Strategy

Our framework employs a two-stage strategy that separates motion estimation from reconstruction. The first stage establishes coherent motion estimates (16-19 PSNR) without optimizing Gaussian points directly. Control points define the deformation field but don't serve as Gaussian centers. Gaussians are initialized independently during the second stage, guided by the established motion field, enabling more stable convergence and higher-quality results.

### F.4 Control Point Efficiency

Our control point approach delivers two significant advantages: (1) a 5× improvement in training time by focusing optimization efforts only on dynamic regions; and (2) a remarkably compact representation requiring only 80MB of storage compared to 153-200MB for competing methods. These efficiency gains enable handling of challenging scenes with large motion magnitudes and longer sequences, as demonstrated by superior performance on the DyNeRF dataset.

### F.5 Deformation Modeling

Our approach employs As-Rigid-As-Possible (ARAP) regularization as a flexible prior for both rigid and non-rigid deformations. For soft-body objects, this acts as a smoothness constraint rather than enforcing strict rigidity. The system adaptively allocates more control points to highly deformable regions, creating a finer deformation grid where needed. This approach successfully handles soft objects, as demonstrated in the "peel-banana" sequence.

### F.6 Implementation Parameters

Our implementation uses 512 control points by default, initialized uniformly with higher density in dynamic regions. Performance remains robust across a range of control point quantities (100-1000), reflecting the effectiveness of our adaptive allocation strategy that focuses computational resources based on motion significance rather than a fixed distribution.

