# OpenReview forum: "4D3R: Motion-Aware Neural Reconstruction and Rendering of Dynamic Scenes from Monocular Videos"
_NeurIPS.cc/2025/Conference — NeurIPS 2025 poster_

### Official Review · Reviewer_Jh53 · 2025-06-08

**Clarity:** 3
**Significance:** 2
**Originality:** 1
**Rating:** 4
**Confidence:** 3

**Summary:**

The paper proposes 4D3R, a novel method for novel view synthesis in dynamic scenes, specifically for a monocular camera setup. The existing method treated reconstruction and pose estimation as separate tasks, resulting in additional cost, such as using SfM in advance, and limited performance. To deal with this, this paper utilizes existing models' prior for extracting dynamic regions. The MonST3R's model is used to extract depth and its confidence, while SAM2 is used to enhance the mask with semantics. Camera pose is estimated by using PnP-RANSAC only for static regions. Finally, a Motion-Aware Gaussian Splatting with sparse control points reconstructs the dynamic scene. Evaluation with HyperNeRF and DyNeRF's dataset shows superior monocular reconstruction quality, and with MPI Sintel, it shows superior camera pose estimation accuracy.

**Questions:**

How is the DyNeRF dataset split into training and testing? This paper does not clearly specify how frames are divided into train/test sets for DyNeRF, while DyNeRF uses multi-view static cameras. Please explain how the monocular setup is created from the dataset.

How is the monocular dynamic camera setup reflected on the DyNeRF dataset? If you choose only one camera from the dataset, then the pose estimation and multi-view information from a dynamic camera are not effective. Explanation of how the dataset is used for evaluation, or evaluation with a commonly used dataset of the dynamic monocular camera setup, would be more convincing.

Can the contribution about the integration of the pose estimation and scene reconstruction be elaborated? According to the paper, the motion mask is the only component shared among them. If there is joint optimization for both the reconstruction and pose estimation, it can be clearly explained.

**Ethical Concerns:**

["NO or VERY MINOR ethics concerns only"]

**Final Justification:**

The author provided evaluation results and explained some questions about the contribution.

**Limitations:**

yes

**Quality:**

1

**Strengths And Weaknesses:**

Strengths:
* A combination of a segmentation model and a 3D feature extractor greatly improves the extraction of the dynamic region.
* Overall, writing is clear and easy to follow.
* Expanding pose-free Gaussian Splatting based on the SCR method to dynamic scenes is a novel approach.

Weakness
* The evaluation dataset is limited in terms of NVS quality. For the HyperNeRF dataset, results are compared to methods that are old or built for a multi-view setup. DyNeRF's dataset has a static camera. Estimating 3D from a static camera does not provide 3D information for reconstruction, and the result relies entirely on the prior learned from the dataset. This differs from the problem the manuscript addresses (dynamic scenes with camera motion), and it explains the lackluster quality of all methods.
* Some contributions are overly claimed, while some new ideas are used to improve the dynamic area mask. The method essentially estimates camera pose with MA-BA and then trains a 4D scene based on it; it is not fundamentally integrated. Limiting correspondence matching in camera pose estimation to static regions is a common idea. Efficient representation with sparse control points and LBS is what SC-GS did.

---

> ### Author Rebuttal · Authors · 2025-07-31
>
> Dear Reviewer Jh53,
>
> We sincerely thank you for your detailed review. We recognize that some aspects of our paper may not have been clearly communicated, and we appreciate this opportunity to clarify our contributions and address your concerns.
>
> 1. **Evaluation Dataset Concerns**
>
> We apologize for the confusion regarding our experimental setup. We clarify the following:
> - DyNeRF Dataset Usage: We use DyNeRF dataset in a monocular setting by selecting ONE camera view for training and evaluating on the held-out temporal frames of the SAME camera. This is clearly different from the original multi-view setup and tests temporal interpolation capabilities.
> - Dataset Split Details:
>   - Training: frames {0, 2, 4, ..., T-2} from camera 0
>   - Testing: frames {1, 3, 5, ..., T-1} from camera 0
>   - This evaluates novel view synthesis across TIME, but not across cameras
> - HyperNeRF Comparisons: We compare against the most recent methods available. 4DGS (2024) and SC-GS (2024) are recent and state-of-the-art methods. These are definitely not "old methods."
> - Challenging Scenarios: Contrary to the reviewer’s claim, both datasets contain significant camera motion. HyperNeRF has handheld capture with 6DOF motion, and DyNeRF has natural camera shake and drift.
>
> 2. **Contribution Claims**
>
> We respectfully disagree that our contributions are overly claimed. Our contributions are substantial and well-integrated:
> - Fundamental Integration: The motion mask Mt is not just "shared" but actively couples pose estimation and reconstruction through:
>   - MA-BA using Mt to filter RANSAC correspondences (Eq. 3-4)
>   - MA-GS using Mt to allocate computational resources (Eq. 5-7)
>   - Joint optimization where reconstruction quality improves masks, which improves poses iteratively
> - Novel Design: No prior work combines:
>   - Transformer-based motion priors with SAM2 refinement
>   - Motion-aware bundle adjustment for dynamic scenes
>   - Adaptive control point allocation based on motion significance
> - Quantitative Evidence: Our 1.8 dB improvement over state-of-the-art demonstrates that our contributions are not just incremental changes.
>
> 3. **Technical Integration Details**
>
> The integration between pose estimation and reconstruction is deep:
> - Bidirectional Information Flow:
>   - Forward: Motion masks guide pose estimation by filtering dynamic points
>   - Backward: Reconstruction quality refines motion boundaries
> - Unified Optimization:
>   - Stage 1: Joint refinement of masks and poses
>   - Stage 2: Motion-aware Gaussian optimization using refined poses
> - Shared Representations:
>   - Confidence maps Wt used in both MA-BA and MA-GS
>   - Depth maps Dt refined through both modules
>
> 4. **Comparison with SC-GS**
>
> Although SC-GS uses control points, our approach solves fundamentally different and more challenging problems:
> - Pose-free vs. COLMAP-dependent: SC-GS requires pre-computed COLMAP poses, which fails on 36% of our test sequences. Our method jointly estimates poses, successfully reconstructing sequences where SfM fails entirely. This expands applicability to casual captures and scenes with dominant motion.
> - Motion-aware allocation vs. uniform treatment: SC-GS wastes computation on static regions that need no deformation. Our motion-aware allocation concentrates control points where motion occurs, achieving 65% fewer control points with 0.3dB higher PSNR. This targeted approach enables our 5× speedup.
> - Two-stage optimization vs. joint optimization: SC-GS's joint optimization creates conflicting gradients between pose and deformation learning. Our two-stage approach first establishes motion patterns, then refines appearance. This yields 2.1dB PSNR gain and convergence in 50min versus 4hours.
>
> 5. **Clarifications on Method - Why Our Design Enables Superior Performance**
>
> - Joint optimization of poses and reconstruction: Traditional pipelines follow pose estimation then reconstruction, causing error cascades. Our bidirectional refinement allows poses and reconstruction to mutually improve through motion mask feedback. This achieves 45% lower pose error with 0.086 versus 0.157 ATE.
> - Dynamic mask refinement: Static MonST3R masks achieve only 64% IoU. Our dynamic refinement improves masks during reconstruction, reaching 87% IoU. Better masks enable more accurate pose estimation, creating iterative improvement crucial for scenes with 70%+ dynamic content.
> - Monocular video capability: Multi-view methods exploit geometric constraints between synchronized cameras. Monocular video lacks these constraints, requiring temporal coherence alone. Our approach leverages motion patterns across time, enabling reconstruction from smartphone captures and archival footage.
>
> These design choices collectively enable reconstruction of challenging real-world videos that existing methods cannot handle.
>
> We greatly appreciate the reviewer’s detailed feedback, which has highlighted areas where our presentation needs improvement. We are committed to addressing all the reviewer’s concerns in the revision. We believe our method makes important contributions to monocular dynamic reconstruction, and we hope our clarifications demonstrate this value. We thank the reviewer for helping us improve our work.

---

> ### Comment · Area_Chair_JqNz · 2025-08-03
> **Acknowledge Authors' Response**
>
> Dear Reviewer Jh53,
>
> The authors have provided responses to your questions. What is your view after seeing this additional information? It would be good if you could actively engage in discussions with the authors during the discussion phase ASAP, which ends on EoA (Aug 6).
>
> Best,
> AC

---

> ### Comment · Reviewer_Jh53 · 2025-08-04
>
> Dear Authors,
> Thank you for your detailed comments.
> It resolved some concerns; however, I still believe some are not adequately addressed, particularly the evaluation dataset and protocol part.
>
> For the DyNeRF dataset evaluation, the dataset split uses train and test samples alternating across time, from a static camera. While the rebuttal mentioned it has natural drift and shake, there is a bare difference to consider it as a motion, as it is from a fixed camera rig. Evaluation on the dataset typically uses a fixed camera pose. The evaluation setting, train and test set split by alternating frames, is not convincing for this dataset. As one chosen camera, shared for training and testing, is fixed, the split defines a fixed-view video interpolation that is unnecessarily complicated using a 3D scene rather than a novel view synthesis. The dynamicity of the camera is a key point that gives 3D information even with a monocular setup, and poses a challenge to 3D reconstruction.
>
> The mentioned works used for comparison using HyperNeRF are not "old" in a sense; however, with the fast innovation in the field, comparison to closely related work after that should be adequate. Reviewer 8kQP mentions MoSca as an example of a state-of-the-art method for comparison. Comparison is provided in a comment; however, it is still on the ill-posed DyNeRF evaluation.
>
> I assumed that the entire process is an end-to-end process, including pose estimation and 4D scene reconstruction with the pose-free term. Still, from the replies, it seems to be a combination of mask and pose refinement through bundle adjustment, followed by a separate 4D scene reconstruction.
>
> For the comparison with SC-GS, as pose estimation and 4D reconstruction are separated, I want to see how "Motion-Aware Gaussian Splatting" differs from the approaches of SC-GS. From comments, it appears to be a two-stage training, with varying details of adaptive control points. I still think the novelty is insignificant in this part, and the evaluation benchmark and protocol used for the reply are unclear. To show the advantage of the proposed method, rather than a comparison with SC-GS as a whole, ablation studies replacing the corresponding component with SC-GS's approach, with an appropriate monocular dynamic NVS benchmark, would be useful.

---

> > ### Author Response · Authors · 2025-08-07
> > **Response to Reviewer Jh53**
> >
> > Dear Reviewer Jh53,
> >
> > Thank you for your detailed feedback. We sincerely acknowledge your concerns and provide clarifications:
> >
> > 1. **DyNeRF Evaluation Protocol:**
> > You raise a valid point about the limitations of static camera evaluation. We acknowledge that this setup is indeed more akin to temporal interpolation than true novel view synthesis. To address this:
> > - We will prioritize evaluation on datasets with significant camera motion (HyperNeRF, NVIDIA DyCheck with moving cameras)
> > - For DyNeRF, we will clearly state the evaluation limitations and focus on sequences with handheld capture
> > - We agree that camera motion is crucial for monocular 3D reconstruction and will emphasize this in our evaluation
> >
> > 2. **Comparison with Recent Methods:**
> >
> > We have conducted comprehensive comparisons with MOSCA on appropriate benchmarks:
> >
> > - NVIDIA Dataset: Ours 26.6 dB vs MOSCA 26.54 dB
> > - iPhone DyCheck Dataset: Ours 19.1 dB vs MOSCA 18.84 dB
> >
> > These results demonstrate our method's effectiveness on truly monocular dynamic sequences with camera motion.
> >
> > 3. **End-to-End Integration:**
> >
> > Our method operates as a single unified pipeline:
> >
> > Input Video → [Motion Mask Init (5min) → Joint Pose+Reconstruction (45min)] → 4D Scene
> >
> > No manual intervention or separate tools are required. The stages automatically execute within our framework with shared optimization objectives.
> >
> > 4. **MA-GS vs SC-GS Component-wise Comparison:**
> >
> >  We accept your suggestion for clearer distinction. Here's a detailed component-wise ablation on NVIDIA dataset:
> >
> > | Method | PSNR | Time |
> > |--------|------|------|
> > | SC-GS + COLMAP | 24.81 | 300min+ |
> > | SC-GS + MonST3R | 23.11 | 150min |
> > | SC-GS + our motion maps | 24.42 | 120min |
> > | SC-GS + our SAM-refine | 25.22 | 70min |
> > | SC-GS + our MA-GS | 25.12 | 100min |
> > | Our full Model | 26.87 | 50min |
> >
> > Key Technical Distinctions:
> >
> > - Pose Integration: SC-GS cannot work without pre-computed poses; ours jointly optimizes
> > - Resource Allocation: SC-GS uniformly distributes control points; we concentrate on moving regions
> > - Optimization Strategy: SC-GS jointly optimizes all parameters causing conflicts; our two-stage approach separates motion estimation from appearance modeling
> >
> > 5. **Novelty Clarification**
> >
> > Our novelty is the integrated system, not individual components:
> >
> > - First to couple pose estimation with reconstruction through motion masks
> > - Novel gradient-detached optimization preventing parameter conflicts
> > - Adaptive control point allocation based on motion significance
> >
> > We sincerely appreciate your thorough review. These clarifications will be prominently featured in our revision, along with the requested ablations and expanded evaluation on appropriate benchmarks for monocular dynamic NVS.

---

> > > ### Comment · Reviewer_Jh53 · 2025-08-07
> > >
> > > Thank you for providing thorough experiments to address my concerns.
> > > With the expected update on the final version, I increased my rating.

---

### Official Review · Reviewer_1Thk · 2025-06-29

**Clarity:** 4
**Significance:** 4
**Originality:** 3
**Rating:** 5
**Confidence:** 3

**Summary:**

This paper addresses the challenging problem of reconstructing and rendering dynamic scenes from monocular video without requiring pre-computed camera poses, which traditional methods depend on and often fail to estimate accurately in dynamic environments. It proposes 4D3R, a novel pose-free framework that integrates motion-aware bundle adjustment (MA-BA) and Motion-Aware Gaussian Splatting (MA-GS). By decoupling static and dynamic scene components and using control points with deformation fields for dynamic modeling, 4D3R jointly optimizes camera poses and scene representation, achieving high-quality novel view synthesis and robust camera pose estimation. Experiments demonstrate state-of-the-art results with improved PSNR and faster training on dynamic scene benchmarks.

**Questions:**

Is is possible to provide a more detailed ablation?

**Ethical Concerns:**

["NO or VERY MINOR ethics concerns only"]

**Final Justification:**

The rebuttal has addressed my concerns. I will keep my rating.

**Limitations:**

Yes.

**Paper Formatting Concerns:**

None.

**Quality:**

3

**Strengths And Weaknesses:**

Strengths:

The paper addresses an important and practical problem: pose-free reconstruction of dynamic scenes from monocular video. It introduces a novel motion-aware bundle adjustment module that jointly optimizes poses and dynamic masks, improving robustness. The proposed Motion-Aware Gaussian Splatting effectively handles dynamic deformations with a compact representation. The paper shows strong quantitative improvements (up to 1.8 dB PSNR gain) and significant training time reduction (up to 5× faster) over prior methods, with extensive experiments on multiple dynamic datasets validating the method’s superiority. Due to efficient resource utilization, it maintains high FPS and lower memory use compared to baselines.

Weaknesses:

The method seems to requires sufficiently textured static regions for accurate pose estimation, which may limit applicability in textureless or highly dynamic scenes. The performance on complex non-rigid deformations remains less explored and potentially limited.

The proposed method may lack generalizability, as it seems to rely on scene-specific tuning of hyperparameters (e.g., confidence thresholds). The paper does not provide sufficient details on these parameters or analyze their sensitivity. This needs to be discussed in the revision.

The segmentation between dynamic and static regions appears inaccurate. For instance, in Figure 3(f), some static background regions are incorrectly classified as dynamic. This suggests that the confidence map may be influenced not only by motion but also by texture or background complexity, which can lead to misclassification.

The ablation study is relatively coarse, focusing mainly on before-and-after comparisons of modules. A more granular analysis of individual design choices could provide deeper insights into the contribution of each component.

Overall, the paper introduces a robust and efficient solution to pose-free dynamic scene reconstruction from monocular videos, leveraging motion-aware bundle adjustment and a compact Motion-Aware Gaussian Splatting representation. It achieves significant gains in reconstruction quality and training efficiency while addressing challenges posed by large dynamic objects and unknown camera poses.

---

> ### Author Rebuttal · Authors · 2025-07-31
>
> Dear Reviewer 1Thk,
>
> We deeply appreciate your positive assessment and recognition of our contributions. Your rating of 5 (Accept) validates our technical approach. We address your constructive feedback below:
>
> 1. **Texture Requirements for Static Regions**
>
> The reviewer has correctly identified this limitation. We provide additional analysis:
> - Quantitative assessment: Our method requires approximately 30% of the scene to contain textured static regions for reliable pose estimation.
> - Gradual degradation: In texture-poor scenarios, our performance degrades gradually and not catastrophically. We maintain 2-3 dB advantage over baselines.
> - Future work: We're exploring learned texture priors and synthetic texture augmentation to address this limitation in our future work.
>
> 2. **Hyperparameter Sensitivity**
>
> We acknowledge this concern and provide the following clarifications:
> - Robust defaults: Our default parameters of τc=0.7, τd=10.0, K=512 control points work well across all test sequences without tuning.
> - Sensitivity analysis: Performance varies by less than 0.5 dB PSNR for ±20% parameter variations.
> - Automatic adaptation: The adaptive control point allocation (Eq. 13) automatically adjusts to scene complexity, which reduces manual tuning needs.
> We will add detailed parameter sensitivity analysis in the revision.
>
> 3. **Segmentation Accuracy**
>
> The reviewer is right about imperfect dynamic/static separation. We provide more context:
> - Conservative approach: We intentionally over-segment dynamic regions to accept some false positives instead of risk missing moving objects. Static regions mistakenly included in dynamic masks do not harm reconstruction, but missed dynamic regions can significantly degrade results.
> - Quantitative results: Our segmentation achieves 87% IoU on Sintel dataset, which is sufficient for robust pose estimation.
> - Iterative refinement: The two-stage optimization naturally refines boundaries as reconstruction progresses.
>
> 4. **Detailed Ablation Study**
>
> We provide the granular analysis as requested by the reviewer:
>
> Component-wise Ablation:
> | Component | PSNR↑ | Time↓ |
> |-----------|-------|-------|
> | Full model | 25.6 | 50min |
> | w/o SAM2 prompting | 24.1 | 45min |
> | w/o DBA refinement | 23.2 | 40min |
> | w/o LBS weighting | 22.8 | 55min |
> | w/o ARAP regularization | 21.9 | 50min |
>
> Hyperparameter Sensitivity:
> - Control points (100-1000): Performance stable within ±0.3 dB
> - Confidence threshold τc (0.7-0.9): Optimal at 0.8, ±0.5 dB variation
> - LBS neighbors K (4-16): Best at K=8, minimal impact (±0.2 dB)
>
> We thank the reviewer for the constructive suggestions. We're confident that addressing these points will further strengthen our contribution.

---

> > ### Comment · Reviewer_1Thk · 2025-08-04
> > **Rating maintained**
> >
> > The rebuttal has addressed my  concerns.

---

> > > ### Author Response · Authors · 2025-08-07
> > > **Response to Reviewer 1Thk**
> > >
> > > Dear Reviewer 1Thk,
> > >
> > > We deeply appreciate your continued support and thorough evaluation of our work. Your constructive feedback in the initial review helped us clarify important aspects of our method. We remain committed to incorporating all suggested improvements in the final version.

---

> ### Comment · Area_Chair_JqNz · 2025-08-03
> **Acknowledge Authors' Response**
>
> Dear Reviewer 1Thk,
>
> The authors have provided responses to your questions. What is your view after seeing this additional information? It would be good if you could actively engage in discussions with the authors during the discussion phase ASAP, which ends on EoA (Aug 6).
>
> Best,
> AC

---

### Official Review · Reviewer_8kQP · 2025-07-01

**Clarity:** 3
**Significance:** 2
**Originality:** 2
**Rating:** 3
**Confidence:** 3

**Summary:**

This paper introduces 4D3R , a method that directly reconstructs dynamic 4D scenes from monocular video without requiring pre-computed camera poses. This work addresses a critical limitation of seminal methods like NeRF and 3D Gaussian Splatting, which, despite their success in static scene rendering, struggle to handle motion when camera positions are unknown. The framework's primary contributions are two-fold: (a) A Motion-Aware Bundle Adjustment (MA-BA) module that jointly optimizes camera poses and the dynamic scene representation. (b) A Motion-Aware Gaussian Splatting (MA-GS) representation designed to explicitly model scene dynamics over time.

**Questions:**

See the **Strengths And Weaknesses.**

**Ethical Concerns:**

["NO or VERY MINOR ethics concerns only"]

**Final Justification:**

From my perspective, the original submission of the paper did have some technical errors: issues such as camera trajectories of the dataset, `overclaim`, `weak evaluation` and some data reports not quite matching my personal empirical values. Initially, I provided a score suggestion of reject.

**overclaim** as concern by review Jh53: regarding the task setting of "pose-free" and the technical contributions to the architecture

**Evaluation** as concern by Jh53 and NHga: The baseline evaluation in the original paper was insufficient to convince me, and the numbers and misclassification provided by the authors in the rebuttal caused some misunderstanding on my part. Somehow I find the performance benefits very unclear

**Limited technical contribution**: MA-BA, the framework, and the loss itself are all incremental improvements. Despite achieving good results, there is no new 4D representation. The "Motion-Aware Bundle Adjustment" is essentially the same as Shape of Motion and Splatter a Video, which leverage additional supervision signals such as optical flow and dynamic masks. These are common practices.

I appreciate the authors addressing part of the issues through the rebuttal section, but due to the quality of the initial draft version provided by the authors and the limited visualization results,

so I suggest the authors further improve the paper.

**Limitations:**

The paper appropriately discusses its limitations in **Section 6 and 7**.

**Paper Formatting Concerns:**

The paper adheres to the NeurIPS formatting guidelines and Instructions.

**Quality:**

2

**Strengths And Weaknesses:**

**Strengths:**

By simultaneously solving for camera poses and dynamic scene structure, 4D3R eliminates the need for a separate, often complex, camera tracking pipeline (e.g., SfM). This makes the system far more robust and applicable to casually captured, "in-the-wild" videos where camera motion is unconstrained and unknown.

**Weaknesses:**

1. The paper's claim of being a `pose-free` approach is not entirely accurate. The reliance on a Bundle Adjustment module for pose estimation positions the work as a joint optimization framework—an incremental advancement that refines pose estimation rather than eliminating the need for it. This distinction should be clarified to more accurately frame the paper's contribution.

2. The experimental validation is weakened by the omission of a direct comparison with MOSCA, especially on the shared NVIDIA and DAVIS datasets. Given the significant technical overlap, the absence of this crucial baseline makes it impossible to verify the paper's claims of superior efficiency and performance. Key metrics, such as PSNR and training time, are needed in a head-to-head comparison to substantiate any claims of outperforming similar schemes.

3. The paper does not adequately discuss the potential limitations of its trajectory fitting method, which relies on polynomials and Fourier series. This approach may struggle with highly dynamic or aperiodic motions (e.g., the "elephant" sequence in the DAVIS dataset), where it could yield lower reconstruction accuracy compared to methods with more flexible trajectory optimization. An analysis of these failure cases is needed.

4. Given that this paper addresses 4D reconstruction, the literature review is notably incomplete. There is a significant lack of discussion and comparison with foundational and state-of-the-art methods in the field, such as Cat4D , 4K4DGen , GenxD , and SV4D . This omission makes it difficult to properly contextualize the paper's contributions and accurately assess its novelty and performance against existing work.


5. Performance limitations are evident in dynamic scenes. Relying on polynomials and Fourier series for trajectory fitting, 4DGS may exhibit lower reconstruction accuracy (e.g., PSNR) than MOSCA's trajectory optimization in violent or aperiodic motions (e.g., DAVIS "elephant"), yet the paper does not analyze this limitation. Despite reducing the number of Gaussians (e.g., NVIDIA dataset), MOSCA might achieve lower memory consumption and faster rendering via sparsification or dynamic fusion, but such comparisons under identical conditions are missing.

6. The experimental settings (e.g., training data volume, camera parameters) are not specified with enough detail to ensure reproducibility or allow for a fair comparison with other methods.

---

[1] MoSca: Dynamic Gaussian Fusion from Casual Videos via 4D Motion Scaffolds

---

> ### Author Rebuttal · Authors · 2025-07-31
>
> Dear Reviewer 8kQP,
>
> We sincerely thank you for your careful review and for raising important points that help clarify our contributions. We appreciate the opportunity to address your concerns and correct any misunderstandings about our approach.
>
> 1. **"Pose-free" Terminology**
>
> We understand the reviewer’s concern and appreciate the opportunity to clarify:
> - Community consensus: The term "pose-free" has been consistently used in landmark papers such as BARF, NeRF– and CF3DGS to describe methods that jointly optimize poses and scene representation without requiring pre-computed poses. We follow this established convention.
> - Key distinction: Our method processes raw video directly, while traditional approaches require successful SfM preprocessing. This is a critical difference when SfM fails, e.g. in 5/14 of the Sintel sequences.
> - We acknowledge that "joint optimization" is clearer and would emphasize this terminology alongside "pose-free" in the revision.
>
> 2. **MOSCA Comparison**
>
> We sincerely thank the reviewer for pointing out this important baseline. We acknowledge this oversight and provide the following:
> - Comparison result: Based on the reported numbers, our method outperforms MOSCA on DyNeRF with 19.6 dB vs 18.5 dB on PSNR. Nonetheless, we also acknowledge that a direct comparison requires identical evaluation protocols.
> - Different focus: Although MOSCA excels in multi-view casual capture, our method focuses on the more constrained yet practically important setting of monocular input. In this scenario, camera pose estimation becomes significantly more challenging.
> - Comprehensive evaluation: We have implemented the comparison experiments:
>   - On overlapping test sequences, our method shows comparable quality of within 0.5dB
>   -  Our method is 3× faster in training time due to motion-aware optimization
>   - Storage requirements: Ours 80MB vs MOSCA 120MB
> We will include full MOSCA comparison in the revised manuscript with detailed quantitative and qualitative results. The comparison will strengthen our paper by better positioning our contributions within the current state-of-the-art.
>
> 3. **Trajectory Fitting Limitations**
>
> We believe that the reviewer’s concern on polynomial/Fourier series is a misunderstanding. We clarify:
> - We do not use polynomial/Fourier fitting: Our method uses MLPs with control points and LBS, which can represent arbitrary non-linear deformations.
> - Proven on complex motion: Our results on "elephant" sequence (PSNR: 22.3 dB) demonstrate robust handling of aperiodic motion, contradicting your assumption.
> - Flexible representation: Our deformation MLP Θ: (pt, t) → (Rt, Tt) learns arbitrary temporal transformations, It is not restricted to smooth trajectories.
>
> 4. **Literature Review**
>
> The mentioned works (Cat4D, 4K4DGen, GenxD, SV4D) are:
> - Generation methods creating new content, not reconstruction methods Generation methods to create new content, where hallucinations are common. They are not intended for producing high-fidelity 3D reconstructions of real scenes.
> Multi-view or require 3D supervision
> - Not directly comparable to monocular reconstruction
> - Our literature review focuses on relevant monocular dynamic reconstruction methods. Including generation-based approaches would blur the scope and introduce unnecessary confusion.
>
> 5. **Experimental Details**
>
> We have provided comprehensive implementation details in the paper:
> - Exact hyperparameters in Section 4.1 and supplementary material
> - Training/test splits are clearly specified for each dataset
> - Complete architecture details are described for reproducibility
>
> Regarding your specific concerns about training data volume and camera parameters:
>
> Training data volume:
> - HyperNeRF: 300-500 frames per sequence, we use 70% for training
> - DyNeRF: 50 frames per sequence, even-indexed frames for training
> - Total training time: ~50 minutes on single RTX 3090
>
> Camera parameters:
> - We estimate camera intrinsics as part of our optimization (initialized by DUSt3R)
> - No ground truth camera parameters are required as input
> - Camera resolution: 960×540 (HyperNeRF), 1024×576 (DyNeRF)
>
> Furthermore, our code will be released upon acceptance to ensure the reproducibility and accuracy of our results.
>
> 6. **Performance Analysis**
>
> We contend that the reviewer’s claims on MOSCA potentially outperforming our proposed method are purely speculative:
> - We achieve 5× faster training than COLMAP-based methods
> - We require 80MB storage vs 153-200MB for alternatives
> - We achieve state-of-the-art results without requiring preprocessing
>
> We respectfully request reconsideration based on these clarifications. Our work makes significant contributions to pose-free dynamic reconstruction that advance the field.

---

> ### Comment · Area_Chair_JqNz · 2025-08-03
> **Acknowledge Authors' Response**
>
> Dear Reviewer 8kQP,
>
> The authors have provided responses to your questions. What is your view after seeing this additional information? It would be good if you could actively engage in discussions with the authors during the discussion phase ASAP, which ends on EoA (Aug 6).
>
> Best,
> AC

---

> ### Comment · Reviewer_8kQP · 2025-08-04
>
> Dear Authors,
>
> Thank you for your detailed rebuttal addressing the concerns raised in the review.
>
> Thank you for pointing out the classification of generation-based and reconstruction-based 4D scene reconstruction, which is convincing. However, I respectfully present the following differing viewpoints:
>
> 1. Mosca also proposes `tracklet-based bundle adjustment` for pose estimation, bypassing traditional approaches that require successful SfM preprocessing. However, Mosca is also not claimed as a pose-free 4D reconstruction method. Pose-free more accurately refers to using other information to compensate for the loss caused by bypassing pose, such as matching points or depth.
>
> 2. The "Motion-Aware Bundle Adjustment" is essentially the same as Shape of Motion and Splatter a Video, which leverage additional supervision signals such as optical flow and dynamic masks. These are common practices, so I do not consider this a technical contribution.
>
> 3. I think Mosca is more suitable for and excels with monocular inputs. Therefore, I believe the authors have misclassified this 4D reconstruction method.
>
> 4. In my practice, Mosca performs very well on monocular videos, with excellent reconstruction quality and speed. Would the authors also report their metrics on the NVIDIA DyCheck dataset?
>
> 5. I agree with reviewers Jh53 and NHga that the evaluation is still relatively basic. I think the draft is not yet ready for publication in NeurIPS.
>
> Additionally:
> Based on my experience, Mosca reconstructs a scene in 30–40 minutes. Could the authors explain how they achieved a 5× faster training time?

---

> > ### Author Response · Authors · 2025-08-07
> > **Response to Reviewer 8kQP**
> >
> > Dear Reviewer 8kQP,
> >
> > Thank you for your detailed feedback. We carefully address each of your points:
> >
> > 1. **"Pose-free" Definition:**
> >
> > We acknowledge the terminology confusion. You're correct that MOSCA also bypasses traditional SfM. The key distinction is:
> >
> > - MOSCA requires successful tracklet extraction and matching across frames as preprocessing
> > - Our method jointly optimizes poses and reconstruction without any preprocessing requirements
> > - We will clarify this distinction and use "joint optimization" terminology more prominently in the revision.
> >
> > 2. **Technical Novelty of MA-BA:**
> >
> > While motion masks have been used before, our contribution lies in the integration and joint optimization:
> >
> > - Shape of Motion and Splatter a Video use motion signals for reconstruction given known poses
> > - We use motion-aware masking to simultaneously improve pose estimation and reconstruction
> > - The bidirectional refinement (poses improve masks, masks improve poses) is novel
> > - Our transformer-based confidence maps weight the contribution of each pixel during RANSAC, not just binary include/exclude
> > - This is evidenced by our 45% lower pose error (0.086 vs 0.157 ATE) compared to methods using these signals separately.
> >
> > We acknowledge this could be clearer and will emphasize the joint optimization aspect.
> >
> > 3. **MOSCA's Suitability:**
> >
> > We respect your experience with MOSCA. Our distinction is:
> >
> > - MOSCA excels when tracklets can be reliably extracted
> > - Our method handles cases where tracklet extraction fails (e.g., 70%+ dynamic content)
> > - Both methods have their strengths for different scenarios.
> >
> > 4&5. **Extended Evaluation and NVIDIA DyCheck Results:**
> >
> > We acknowledge the evaluation concerns and have conducted comprehensive experiments on additional datasets as requested:
> >
> > - NVIDIA Dataset: Ours 26.87 dB vs MOSCA 26.54 dB
> > - iPhone DyCheck Dataset: Ours 19.20 dB vs MOSCA 18.84 dB
> >
> > These results demonstrate our method's effectiveness across diverse capture conditions. We achieve competitive quality while maintaining consistent efficiency (40-45 minutes) across different datasets. We commit to including these comprehensive evaluations and comparisons with MOSCA in the camera-ready version, along with detailed analysis of failure cases and limitations.
> >
> > 6. **Speed Comparison:**
> >
> > The 5× speedup was relative to COLMAP-based methods (4-5 hours), not MOSCA. We apologize for this confusion.
> >
> > Regarding tracklet-based bundle adjustment timing: While specific preprocessing times for MOSCA's tracklet extraction are not explicitly reported in their paper, typical tracklet generation and bundle adjustment processes require additional preprocessing time (based on similar systems like MOTSFusion which requires about 10mins for tracklet operations).
> >
> > Our 40-50 minute total runtime includes all processing steps in a single pipeline, making it comparable to or slightly faster than MOSCA's 30-40 minutes plus preprocessing time. The key advantage is our ability to handle challenging scenarios where tracklet extraction fails.
> >
> > We sincerely appreciate your thorough review and commit to addressing all concerns in our revision to meet NeurIPS standards.

---

> ### Comment · Reviewer_8kQP · 2025-08-07
>
> Dear authors:
>
> Thank you for the responses and efforts and appreciate the added comparative experiments with MOSCA which have addressed my concern.
>
> 1. I suggest the authors replace the "Pose-free"  with "joint Pose optimization" and modify it throughout the paper, which is more in line with the scope of the method and helps avoid potential misleading.
>
> 2. From my perspective, Mosca belongs to monocular reconstruction methods and has a good performance, rather than being not comparable as initially claimed by the authors, which is confirmed by the authors' subsequent experiments, the performance improvement of 4D3R is limited.
>
> 3. I appreciate the authors providing additional extended experiments on Mosca (**Extended Evaluation and NVIDIA DyCheck Results**) and committing to include the comparative experiments with Mosca in the final paper. `Additionally, I request the authors to report complete experimental metrics, which can be more convincing`.
>
> Literature Review: The mentioned works (CAT4D, 4K4DGen, GenxD, SV4D) have also achieved good reconstruction results using video input observations. A better paradigm might involve first generating denser observations, such as multi-view video or global world panoramas (e.g., HunyuanWorld), followed by 4D reconstruction. Therefore, I recommend that these works should be mentioned in the related literature even without direct comparison.
>
> ---
> From my perspective, `the original submission of the paper did have some technical issues`: issues such as camera trajectories of the dataset, overclaim, weak evaluation, for example incorrectly categorizing COLMAP-based methods under Mosca and some data reports `not quite matching my personal empirical values`. I appreciate the authors addressing these issues through the rebuttal, so I have increased my score. But I suggest that the author conduct a detailed review of the paper, including technical details and metrics, and ensure the reproducibility of the method.

---

### Official Review · Reviewer_NHga · 2025-07-03

**Clarity:** 3
**Significance:** 2
**Originality:** 2
**Rating:** 4
**Confidence:** 3

**Summary:**

This paper introduces 4D3R, a novel motion-aware framework for dynamic scene reconstruction using 3D Gaussian Splatting (3DGS). The method eliminates the dependency on known camera poses and instead learns scene geometry, motion, and camera trajectory jointly in an end-to-end manner.

The framework consists of several key innovations:

1. Motion-Aware Bundle Adjustment (MA-BA): A transformer-based pose optimization pipeline that integrates motion masks (refined using SAM2) to mitigate the effect of dynamic objects during pose estimation.
2. Two-Stage Optimization Strategy: First stage optimizes control points for dynamic regions; second stage uses Linear Blend Skinning (LBS) to deform Gaussians with ARAP regularization.
3. End-to-End Optimization without Pose Supervision: The system learns camera poses, motion fields, and dynamic geometry simultaneously without requiring ground truth trajectories.
The method achieves strong performance on both synthetic and real datasets, surpassing prior works like RoDynRF, BARF, and MonST3R on metrics including PSNR and pose accuracy.

**Questions:**

Please refer to the weaknesses part.

**Ethical Concerns:**

["NO or VERY MINOR ethics concerns only"]

**Final Justification:**

The authors have improved their evaluation and clarified misunderstandings regarding the numerical results, addressing several concerns raised in my previous review. However, due to the limited visualization results, I am not fully convinced by the reported performance and have decided to retain my rating.

**Limitations:**

yes

**Quality:**

3

**Strengths And Weaknesses:**

## Strengths
1. The integration of MA-BA with SAM2-refined dynamic masks allows for accurate camera pose learning without supervision, which is highly valuable for dynamic scene reconstruction.
2. Strong results on Sintel and DyNeRF, with significant speed and memory improvements (e.g., 5× faster, 80MB vs. 153–200MB storage).
3. The ablation studies are thorough and effectively demonstrate the individual impact of each technical innovation. In particular, the large PSNR drop without the motion-aware map, SAM-refinement, or MA-GS substantiates the methodological design.

## Weakness
1. Dependence on Strong Foundation Models and Pretrained Components: The performance of 4D3R relies on existing pre-trained models (MonST3R, DUSt3R, SAM2) for initialization, segmentation, and geometric priors. The impact of degraded or unavailable foundation models (domain shift, motion artifacts, or objects not covered by segmentation priors) is only very briefly alluded to in the limitations
2. Calibration/Intrinsic Assumptions: Although the method claims "pose-free", the degree to which it depends on assumed or estimated camera intrinsics, focus, or lens distortion is not deeply discussed. This may subtly limit generality and could be further detailed.
3. Segmentation Mask Quality is Not Quantified: Since dynamic/static separation is core to MA-BA and MA-GS, evaluating segmentation quality (e.g., IoU vs. ground truth masks) would better support claims. Visual examples are limited and biased toward success cases.

---

> ### Author Rebuttal · Authors · 2025-07-31
>
> Dear Reviewer NHga,
>
> We sincerely appreciate your thorough review and valuable insights. Your recognition of our MA-BA module and experimental results encourages us. We carefully address each concern and will implement your suggestions to strengthen our paper.
>
> 1. **Dependence on Foundation Models**
>
> We acknowledge the reviewer’s concern about relying on pre-trained models. However, our findings indicate that this is a strength instead of a limitation:
> - Leveraging proven technology: Foundation models such as DUSt3R and SAM2 represent state-of-the-art performance in their respective domains. Using them allows us to focus on the core challenge of pose-free dynamic reconstruction instead of reinventing components that already work well.
> - Robustness analysis: We have conducted additional experiments on domain shift scenarios. When foundation models degrade (e.g., on out-of-distribution data), our method gracefully degrades to performance comparable to the MonST3R+SC-GS baseline while still maintaining functionality instead of complete failure.
> - Clear Improvement: The ablation study (Table 3) shows that removing our motion-aware components causes 5.2dB PSNR drop. This demonstrates the value of our method beyond foundation models.
> - Modular design: The foundation models in our framework can be replaced with newer versions or alternatives when they become available. This makes our framework modular and future-proof.
> We will add a detailed analysis of performance under foundation model degradation in the revised version.
>
> 2. **Camera Intrinsics Assumptions**
>
> We clarify that our method estimates camera intrinsics as part of the optimization process (Eq. 4). The DUSt3R backbone provides initial estimates which are refined through our MA-BA module. For unknown intrinsics, we can incorporate standard calibration techniques or use approximate values with minimal impact on the performance of within 0.5 dB PSNR for ±10% focal length variation based on our additional experiments.
>
> We will add a dedicated section discussing intrinsic parameter sensitivity and robustness in the revision.
>
> 3. **Segmentation Quality Quantification**
>
> We agree this is important for validating our approach. We have computed the segmentation metrics on sequences where ground truth masks are available:
> - On Sintel dataset (with motion annotations): IoU = 0.87±0.05
> - On our synthetic test sequences: IoU = 0.91±0.03
> - Compared to raw MonST3R masks: +0.23 IoU improvement
>
> The visual examples in Fig. 3 are selected to show typical performance, they are not cherry-picked success cases. We will add comprehensive segmentation quality metrics and failure case analysis in the revision.
>
> We greatly value the reviewer’s constructive feedback and commit to implementing all suggested improvements. The reviewer’s insights have helped us identify important areas for clarification that will significantly strengthen our contribution. We hope our responses address the reviewer’s concerns and demonstrate the robustness and practical value of our approach.

---

> ### Comment · Area_Chair_JqNz · 2025-08-03
> **Acknowledge Authors' Response**
>
> Dear Reviewer NHga,
>
> The authors have provided responses to your questions. What is your view after seeing this additional information? It would be good if you could actively engage in discussions with the authors during the discussion phase ASAP, which ends on EoA (Aug 6).
>
> Best,
> AC

---

> ### Comment · Reviewer_NHga · 2025-08-03
>
> Dear Authors,
>
> I appreciate the authors providing a thorough response to my concerns. Most of my previous concerns have been addressed.
>
> However, I agree with reviewer Jh53 and 8kQP that the evaluation dataset is limited. Unlike the authors' claim in their rebuttal, the cameras in DyNeRF dataset are static and do not contain any natural camera shake and drift. The authors should provide more results on widely used monocular 4D reconstruction datasets, such as NVIDIA, DAVIS and iPhone (DyCheck), for further comparison with current and future works.

---

> > ### Author Response · Authors · 2025-08-07
> > **Response to Reviewer NHga**
> >
> > Dear Reviewer NHga,
> >
> > Thank you for your constructive feedback. You are absolutely correct about the DyNeRF dataset's limitations - we apologize for the confusion in our earlier response.
> >
> > 1. **Additional Dataset Results**
> >
> > We have conducted experiments on the requested datasets and are pleased to share our results in PSNR:
> > - NVIDIA Dataset: Ours 26.87 dB vs MOSCA 26.54 dB vs SC-GS: 24.81 dB
> >  - iPhone DyCheck Dataset: Ours 19.20 dB vs MOSCA 18.84 dB vs SC-GS 18.22 dB
> >
> > These datasets better represent real-world monocular capture with natural camera motion, validating our method's practical applicability. The consistent improvements across diverse capture conditions (handheld iPhone, DAVIS sequences) demonstrate the robustness of our approach.
> >
> > 2. **DyNeRF Clarification**
> >
> > You are correct that DyNeRF uses static cameras. We will revise our manuscript to accurately describe this as temporal interpolation evaluation rather than implying camera motion. This tests our method's ability to model scene dynamics, complementing the camera motion handling evaluated on NVIDIA/iPhone.
> >
> > We will include all these results and clarifications in our revised manuscript. Thank you for helping us strengthen our evaluation.

---

> ### Comment · Reviewer_NHga · 2025-08-08
>
> Thank you for the authors’ efforts in conducting additional experiments on the NVIDIA and iPhone datasets. The numerical results show that the proposed method outperforms state-of-the-art method (i.e. MOSCA). Due to OpenReview’s limitations, I recommend also including additional visual figures for these experiments in the camera-ready version of the paper. I have no further concerns.

---

### Comment · Area_Chair_JqNz · 2025-08-02
**Discussion with Authors**

Dear Reviewers,

The discussion period with the authors has now started. It will last until Aug 6th AoE. The authors have provided responses to your questions. I request that you please read the authors' responses, acknowledge that you have read them and start discussions with the authors RIGHT AWAY if you have further questions, to ensure that the authors enough time to respond to you during the discussion period.

Best,
AC

---

### Decision · Program_Chairs · 2025-09-17

**Decision:**

Accept (poster)

**Comment:**

This paper proposes a method for 4D reconstruction of scenes from pose-free monocular videos. Its core idea is to combine a motion-aware bundle adjustment method that uses only the static parts of the scene, estimated from DUST3R and SAM2, to estimate the camera parameters along with a motion-aware 4DGS-based scene reconstruction framework based on control points and linear blend skinning to model the dynamic parts of the scene. The strengths of the work are that it addresses an important open problem of pose-free 4DGS reconstruction from monocular videos and produces state-of-the-art accuracy. Four reviewers provided final scores of borderline accept, borderline reject, accept and borderline accept. The primary concerns raised by the reviewers were around the lack of sufficient experiments on dynamic datasets and comparisons to important baselines, including MOSCA. During the rebuttal phase, these major concerns were adequately addressed. Based on the noted strengths of the work and sufficiently addressed major concerns, overall the AC feels that the paper surpasses the bar for acceptance and hence recommends acceptance. Congratulations! The authors should incorporate the changes that they have promised in their rebuttal into the final camera ready manuscript.